# Blink Reflex Examination in Patients with Amyotrophic Lateral Sclerosis Compared to Diseases Affecting the Peripheral Nervous System and Healthy Controls

**DOI:** 10.3390/brainsci13101384

**Published:** 2023-09-28

**Authors:** Róbert Rostás, István Fekete, László Horváth, Klára Fekete

**Affiliations:** 1Division of Radiology and Imaging Science, Department of Medical Imaging, Faculty of Medicine, University of Debrecen, Nagyerdei krt 98, 4032 Debrecen, Hungary; 2Department of Neurology, Faculty of Medicine, University of Debrecen, Móricz Zsigmond krt 22, 4032 Debrecen, Hungary; fekete@med.unideb.hu (I.F.); feketek@med.unideb.hu (K.F.); 3Department of Pharmaceutical Surveillance and Economy, Faculty of Pharmacy, University of Debrecen, Nagyerdei krt 98, 4032 Debrecen, Hungary; lhorvath@med.unideb.hu

**Keywords:** blink reflex, amyotrophic lateral sclerosis, bulbar lesion, upper motor neuron, lower motor neuron, motor neuron disease, peripheral nervous system

## Abstract

Amyotrophic lateral sclerosis (ALS) is a fatal form of neuromuscular disease. The aim of this study was to assess changes in the blink reflex (BR) parameters as a valid and easy-to-use tool in ALS patients. We assessed the BR test in patients with a definitive diagnosis of ALS, healthy volunteers, and patients with diseases affecting the peripheral nervous system. The BR was studied in 29 patients who met the Awaji criteria. Latencies were compared with our healthy controls (*N* = 50) and other diseases of the peripheral nervous system (*N* = 61). The ALS Functional Rating Scale—Revised (ALSFRS-R) was used to evaluate functional status. Significantly prolonged R2i and R2c latencies were found in the ALS group compared with the healthy control group (*p* < 0.001). The latencies of R1, R2i, R2c were all increased in the bulbar subtype compared to the limb-onset subtype (*p* < 0.05). According to our results, BR examination might be a promising tool to monitor the course of the disease or serve as a prognostic biomarker in patients with ALS, but it should be assessed in further studies. The abnormalities detected through BR might help perform earlier interventions in ALS patients and might be useful in other diseases affecting the peripheral nervous system.

## 1. Introduction

Amyotrophic lateral sclerosis (ALS) is the most common incurable motor neuron disease of adult onset, characterized by signs of dysfunction of the lower and upper motor neurons, leading to progressive weakness and atrophy of the bulbar, limb, thoracic, and abdominal muscles [1].

Furthermore, this motor neuron disease (MND) entity is associated with a pseudobulbar effect and, in up to 50% of patients, cognitive and behavioral abnormalities. However, consciousness and vegetative functions are spared [2,3].

The worldwide annual incidence of ALS is about 1.9 per 100,000, and men are affected nearly twice as often as women. The number of affected individuals above 60 years of age is expected to increase. According to epidemiological studies, there will be a 69% increase in the number of patients suffering from ALS in the period from 2015 to 2040 across the globe [1,4].

Since ALS is of notable socioeconomic significance not only for the patients and their caregivers, but also for the healthcare system, it is essential to obtain an early and accurate diagnosis of the disease [5,6]. As making the diagnosis of ALS is a challenging task and requires the exclusion of a lot of other neurological diseases, it is crucial to have supporting diagnostic tools for prognostic purposes, as well as for the physical and medical therapeutic efforts. If the clinical symptoms are suggestive of a motor neuron disease, supportive testing should be carried out. Neurophysiological examinations have been incorporated into the El Escorial criteria and are essential in the diagnosis of ALS, so electromyography (EMG) and nerve conduction studies (NCSs) are the proper diagnostic tools [7,8,9]. Additionally, other neurophysiological tests are available to examine the integrity of the nervous system, including the blink reflex (BR) test [10,11].

BR was initially described for dysfunctions of the brainstem [12]. Activation of Aβ-afferents leads to the early unilateral R1 wave via the primary trigeminal nucleus in the mid pons, facilitated by an oligosynaptic pathway. Aβ- and Aδ-afferent fibers induce the bilaterally late R2 waves [13,14,15,16]. The exact central conduction pathways for bilateral R2 responses remain unclear. R2 components’ afferent fibers descend through the spinal trigeminal tract in the medulla oblongata from the pontine formation, terminating in the spinal trigeminal nucleus. The cerebral cortex and basal ganglia influence the facial nucleus function, impacting R2 wave latency [15,16,17]. Polysynaptic pathways convey ipsilateral and contralateral R2 elements across the reticular formation’s lateral tegmental field, connecting with facial nerve nuclei. Comparative studies analyzing R1 and R2 responses have revealed variations in various neurological conditions [18,19,20,21,22,23,24].

The role of the neurophysiological studies is to confirm the suspicion of ALS and exclude other diseases mimicking symptoms of MND. BR is one of the diagnostic tools, but its role in screening patients is still to be established in cases where ALS is suspected. So far, only a few scientific research articles and analyses concerning BR in ALS have been published, and the reported results are controversial. Currently, we are lacking precise statistical data to clarify the changes in the BR response and the blinking pathway in cases of ALS [10,11,25,26,27].

The aim of this study was to assess changes in the blink reflex (BR) parameters as a valid and easy-to-use tool in ALS patients. We assessed the BR test in patients with a definitive diagnosis of ALS, healthy volunteers, and some patients with diseases affecting the peripheral nervous system that might alter the BR response.

## 2. Materials and Methods

### 2.1. Patients and Data

The inclusion time period was between 1 June 2018 and 31 June 2022 at the Department of Neurology of the University of Debrecen. Our neurophysiology unit is a tertiary center with a catchment area of 600,000 inhabitants in a 90 km radius.

A total of 29 patients were enrolled in the study. All of the patients were recruited after the initial diagnosis if they met the Awaji criteria. Clinically, ALS is defined by clinical or electrophysiological evidence of the presence of LMN and UMN signs in the bulbar region and at least 2 spinal regions, or the presence of LMN and UMN signs in 3 spinal regions [8]. Every patient developed progressive motor impairment within one year. All of the patients had had clinical signs of UMN and LMN dysfunction in three regions revealed by repeated clinical assessments [8]. The definite diagnosis of ALS and willingness to take part in the examinations served as inclusion criteria. Evidence of lower motor neuron dysfunction was also confirmed by EMG in all of the patients included in the study.

From the analysis, we excluded individuals with a history of previous cranial nerve lesions, strokes, or traumatic brain injury, as well as those with non-neurological diseases that might cause any potential central or peripheral nerve impairment. Cases of other types of motor neuron diseases were also regarded as exclusion criteria. All of the patients involved in our study were examined using cranial and cervical high-resolution 3-tesla (3T) magnetic resonance tomography in order to exclude any structural abnormalities of the brain or spinal cord. Cerebrospinal fluid tests and blood tests were also performed to exclude other diseases mimicking ALS in all of the cases.

Amyotrophic Lateral Sclerosis Functional Rating Scale—Revised (ALSFRS-R) scores [28,29] were used to assess functional status. Groups were formed according to the onset symptoms: bulbar onset, or limb onset. The two groups were formed according to their onset symptoms. At the time of the examination, it might have occurred that the patient had additional symptoms clinically or subclinically detected by neurophysiological examinations, while also fulfilling the Awaji criteria. A severe group of 4 patients was examined separately, where at the time of the examination the symptoms were very severe and the onset could not be clarified.

Each of the participants signed an informed consent form to take part in the protocol, which was approved by the local ethics committee (RKEB5036-2018).

### 2.2. Blink Reflex

Neither the healthy controls nor the ALS patients received any medication prior to BR testing.

The subjects examined were reclining in a chair comfortably or lying down in a supine position, in a relaxed state with their eyes open. The examination took place in a quiet, electrically shielded, temperature-regulated room of the neurophysiology laboratory at the Department of Neurology. The BR was assessed using two-channel Nicolet Viking Quest EMG equipment (Nicolet Biomedical Inc., Madison, WI, USA).

The recording and reference electrodes were surface plate electrodes, while the stimulating electrode was a surface electrode on a fixed bar. The active recording electrode was placed laterally to the vertical line of the mid position of pupils, over the inferior part of the orbicularis oculi muscles at the lower eyelid. The reference recording electrode was placed at approximately 3–4 cm from the internal edge of the active electrode, at the side of the nose. Recording and reference electrodes were placed on both sides. A ground electrode was placed around the upper arm.

Supraorbital branches of the trigeminal nerve were stimulated transcutaneously on each side by a cathode placed on the ophthalmic arch, 1–2 cm from the midline at the supraorbital notch. In order to avoid the spread of the stimulation to the contralateral supraorbital nerve, the anode was obliquely rotated laterally and placed higher. We tried to minimize habituation by applying stimuli irregularly, at 0.5 Hz, for 0.2 ms. The examination began with test stimuli in order to avoid discomfort or anxiety about stimulation. The subjects reported no discomfort associated with the stimuli. Stimulation was gradually increased by steps of 1 mA, until a reliable and most stable R1 wave appeared reproducibly. We recorded 8 to 10 responses from each side on average, and we determined the shortest latency. We used individual threshold stimulus intensities in order to analyze the R1, R2i, and R2c latencies and obtain an apparent wave. In the absence of the R1 component, the patient was asked to close their eyes gently, and the stimuli were given in pairs at 5 ms intervals.

In consequence of the various methodological approaches and laboratory environments, as well as the different measuring systems assessed, different laboratories might have accepted disparate standard values for the latencies of R1, R2i, and R2c. Conducting studies of the BR, the authors limited control groups and, as a result, different R1, R2i, and R2c components arose. In view of the above, we also established our laboratory-specific values for the R1 and R2 parameters that could be used in our studies prior to examining subjects with suspected ALS. For this purpose, we enrolled 50 healthy volunteers, aged 22 to 75 years (23 female and 27 male subjects; median age = 50 years), who had no history of any central or peripheral nerve diseases. Additionally, Bell’s paresis (*N* = 27) (left, right), myasthenia gravis (ocular, generalized) (*N* = 9), and diabetic polyneuropathy groups (*N* = 25) were also examined for comparison purposes (total *N* = 61). Since laboratory-specific control R1 and R2 parameters were available, subjects with ALS were examined to obtain their R1, R2i, and R2c components. BR parameters were considered pathological if the value was outside the 95th percentile for healthy controls. The values were as follows: R1 left 12.3 ms, R2i 36.44 ms, R2c 36.65 ms, R1 right 12.28 ms, R2c 36.28 ms, R2c 37.13 ms.

### 2.3. Statistical Analysis

We used the software “GraphPad Prism 8.2.1. Statistical program” and Microsoft Office Excel 2019 for statistical analyses. Normality analyses were conducted; if the distribution was normal, we carried out ANOVA, and if the distribution was not normal, Kruskal–Wallis analysis was performed by multiple comparisons. When comparing two groups, the Mann–Whitney test for non-parametric variables and the *t*-test as a parametric test were conducted. For age comparisons, either the *t*-test or ANOVA was used, and the other parameters were tested with the Kruskal–Wallis or Mann–Whitney tests. Statistical significance was considered if *p* < 0.05.

## 3. Results

### 3.1. Baseline Characteristics

Of the 29 patients enrolled, 12 were females (aged 47 to 74; mean age 65 years) and 17 were males (aged 36 to 84 years; mean age 63 years) (Table 1). In all of the study participants, evidence of lower motor neuron dysfunction was also confirmed by EMG. Large motor unit potentials of increased duration and increased amplitude could be detected in at least one body region in 100% of the patients, while 70% of the patients had polyphasia. All of the patients had evidence of ongoing degeneration (80% had fibrillation potentials, 70% had fasciculation potentials, and 60% had positive sharp waves). Thirty-three percent of the patients with ALS had no bulbar signs on the neurological examination at the time of recruitment. Unfortunately, eight patients died within the study period.

### 3.2. Blink Reflex

In Table 2, the latencies of R1, R2i, and R2c are compared between the groups of ALS (all ALS patients, *N* = 29), Bell’s paresis (right, left), myasthenia gravis, diabetic polyneuropathy, and healthy controls. Figure 1 demonstrates a normal record, Figure 2 a pathological one. The R1 and R2 responses (both ipsi- and contralateral) also show extreme variability in amplitude in the normal populations. They may vary even from one healthy subject to the next; therefore, they are not reliable parameters to be used as an index of any abnormality.

All ALS patients had statistically significantly delayed latencies of ipsilateral and contralateral R2 waves when compared with healthy controls (*p* < 0.001) (Table 2). In contrast, there were no statistically significant differences in R1 responses (*p* (left) = 0.12, *p* (right) = 0.18) in ALS patients compared with the healthy control group (Table 2, Figure 3). The healthy controls were significantly younger, so we analyzed the healthy control subgroup: there were no significant differences in the findings between patients under 65 years of age and those older than 65 years (*p* = 0.2).

In Table 2, the *p*-values are summarized, where the latencies of the R waves in different diseases are compared to healthy subjects. In Bell’s paresis, as can be expected, the latencies of R1, ipsilateral R2i, and contralateral R2c are increased. In myasthenia gravis and diabetic polyneuropathy, all R-wave latencies are longer, but in ALS, only the R2i and R2c values differ significantly. Therefore, the difference between ALS and the other examined diseases was the normal R1 (Table 2). Four patients with ALS were hospitalized at a later stage of the disease. Their contralateral R2 waves had completely disappeared and, therefore, could not be quantified (Figure 2). It is also important to study the absent responses. In ALS patients, the absent R2c responses may show the importance of interneurons.

Of the patients, 6 had bulbar onset and 19 had limb onset, with the period from the onset of the disease to the time of examination being 7 and 6.79 months, respectively (Table 3). If the different subgroups of ALS are examined (bulbar vs. limb), significant differences can be observed in terms of age (*p* = 0.004). The latencies of R1, R2i, R2c were all increased in the bulbar-onset subtype compared to the limb-onset subtype (*p* < 0.05). The four severe cases had worse ALSFRS-R scores, and their disease duration before examination was also significantly longer. (*p* < 0.0001). The patients’ orbicularis oculi and frontalis muscles were examined, without clinical findings. The data are summarized in Table 3 and shown in Figure 4. The bulbar- and limb-onset groups did not show significant differences in ALSFRS-R (Table 3).

If the R latencies of BR are examined in the light of disease duration from onset until examination, a positive correlation can be detected with Spearman’s rank test: ρ = 0.5, *p* = 0.007; ρ = 0.18, *p* = 0.25; ρ = 0.02, *p* = 0.39; ρ = 0.35, *p* = 0.07; ρ = −0.18, *p* = 0.25; and ρ = −0.47, *p* = 0.01 by the latencies of R1 left, R2i left, R2c left, R1 right, R2i right, and R2c right, respectively (Figure 5). Despite significant correlations, the r values are low, so cautious interpretation is required.

Using the ALSFRS-R score, better evaluation of the functional status could be achieved; therefore, we used this as a comparison for the R1, R2i, and R2c latencies. In Figure 6, each patient’s BR parameters are shown according to their ALSFRS-R status: longer R2 parameters were observed, while R1 was the same in both the bulbar-onset (Figure 6A) and limb-onset (Figure 6B) groups. Spearman’s rank correlation was computed to assess the relationship between ALSFRS-R and BR latencies, and there was a negative correlation between the variables. In all patients with ALS, ρ = −0.95, *p* < 0.0001; ρ = −0.61, *p* = 0.001; ρ = −0.82, *p* < 0.0001; ρ = −0.90, *p* < 0.0001; ρ = −0.83, *p* < 0.0001; and ρ = −0.80, *p* < 0.0001 by the latencies of R1 left, R2i left, R2c left, R1 right, R2i right, and R2c right, respectively. Among patients with bulbar onset, ρ = −0.90, *p* = 0.005; ρ = −0.93, *p* = 0.002; ρ = −0.84, *p* = 0.017; ρ = −0.96, *p* = 0.0008; ρ = −0.94, *p* = 0.002; and ρ = −0.97, *p* = 0.0004 by the latencies of R1 left, R2i left, R2c left, R1 right, R2i right, and R2c right, respectively. In patients with limb onset, ρ = −0.99, *p* < 0.0001; ρ = −0.96, *p* < 0.0001; ρ = −0.98, *p* < 0.0001; ρ = −0.98, *p* < 0.0001; ρ = −0.97, *p* < 0.0001; and ρ = −0.97, *p* < 0.0001 by the latencies of R1 left, R2i left, R2c left, R1 right, R2i right, and R2c right, respectively.

The symptoms of ALS are usually asymmetrical. In order to compare our data, subgroups were formed within the limb-onset group as follows: right upper limb, left upper limb, right lower limb, and left lower limb. The sample sizes were not balanced, as this was a real-world scenario. In Figure 7A, R1, R2i, and R2c are shown in the different subgroups. All parameters differed significantly between the right upper limb and right lower limb subgroups (Table 4).

According to our findings, patients with right upper limb onset had the best functional status, while those with right lower limb or bulbar onset had the worst (Figure 7B).

## 4. Discussion

The diagnosis of ALS is still challenging, especially in the earlier stages of the disease. EMG and other neurophysiological studies are important diagnostic methods for all patients who present with clinical signs of ALS [30,31,32,33,34,35]. Usually, blinking and other upper facial movements can be retained until the late stages of ALS, making the blink reflex a potential supportive tool. Additionally, not only the conduction of the reflex arch but also the structures of the affected brainstem can be examined using BR studies. However, there are some studies [10,11,25,26,27] that highlight the importance of BR in the assessment of the pathophysiology of ALS.

Despite the fact that ALS is known among laypeople thanks to the media, our patients with bulbar and limb onset still needed 7 months to be recognized and sent to our laboratory, meaning that time-consuming examinations (e.g., MRI) could be started from that point in time at the earliest. Most of the patients had mild or moderate symptoms in the bulbar- and limb-onset groups, but their functional ALSFRS-R status was already rather moderate, without significant differences between the two groups. The four patients in the severe group unfortunately stayed hidden for a long time.

Analyzing the subgroups of ALS, the patients in the bulbar-onset group were younger and of work-capable age; in the other groups, the age was also not extremely high. That accentuates that this disease effects families, not only the individual, and not only in a physical way, but in a complex social way as well, emphasizing the need for support [5,6].

Compared with the healthy control group, all of our patients with ALS had pathological values of BR and detectable losses of connectivity in the lower brainstem interneurons, although no abnormality was seen using imaging techniques (i.e., cranial and cervical MRI). In an earlier publication, even in a well-defined case of localized Wallenberg’s disease, only with the help of BR could the exact damage be identified and understood [18]. Among our ALS patients, we found six cases where no R2c answers could be detected, despite the longer (but measurable) R2i responses. This finding may also support the functional loss of interneurons. Interneurons play a crucial role in mediating the R2 wave of the BR, as supported by several studies, including neuropathological research that corroborated similar findings [36,37,38]. On the physical examination, 76% of the patients in our study had no bulbar or neurological symptoms related to brainstem damage. Progressive disappearance of the contralateral R2 waves in the late stages of the disease indicates the involvement of both UMN and LMN, as well as that of the brainstem interneurons. This can help us to understand more of the dysfunction of the nervous system in ALS and other neurodegenerative diseases.

Comparing the data of patients with some diseases affecting the peripheral nervous system (Table 2), a “pattern” can help in the evaluation. In ALS, R1 does not differ significantly, while R2i and R2c values do; in myasthenia gravis, all R values are increased, while in Bell’s paresis they differ according to the side. Further examining (Table 2) the correlations between the differences in R latencies between groups, the normality of R1 plays an important role. These abnormalities found in a region examined by BR are responsible for gating of the sensorimotor system, highlighting the importance of the cerebral cortex and basal ganglia [38]. A similar “pattern” to ALS can be seen in stroke, damaging the area and proving the lesion of the central pathway [18]. Of course, some limitations hinder these comparisons. The character of the diseases sometimes already determines the age as well, e.g., in myasthenia gravis, the patients were younger. Nevertheless, the pathological latencies were increased and showed a pattern that could be well distinguished from the “ALS pattern”.

In Figure 6, we demonstrate that the R2 answers’ latency is correlated with ALSFRS-R, making BR examination a possible tool to follow up the course of the disease in a patient. When examining BR responses in the light of disease duration, the correlation has to be interpreted very cautiously. Although the *p*-values might be significant, they are accompanied by a less powerful ρ-value. This might be due to either the small sample size and/or the characteristics of the course of the disease, which might show great interindividual differences.

Trying to understand the pathomechanism of BR in ALS patients, we found an interesting aspect of BR parameters in a peripheral nerve in diabetic polyneuropathy without any sign of cranial nerve dysfunction. The answers’ latencies were higher than in healthy controls, meaning that there are already measurable lesions in the cranial nerves before symptoms occur. Although diabetic mononeuropathy in cranial nerves is well known, polyneuropathy is a rather distinct entity affecting the limbs and the vegetative system [39]. This assumption may be relevant in ALS patients as well, meaning that the abnormalities detected with BR are seen earlier than the clinical symptoms, and might help an earlier intervention in order to maintain a better functional status in ALS patients as well. Of course, this thesis should be proven on a larger sample size with follow-up BR examinations. This calls attention to the need for better care and more precise follow-up [2].

The latencies were the most increased in the bulbar-onset group, but these patients already had bulbar symptoms. Comparing the latencies and functional status (ALSFRS-R) between the subgroups of limb onset, we found significant differences only between the right upper limb and right lower limb subgroups, favoring the former (Table 4 and Figure 7). This is an interesting finding, but it might also have occurred by chance due to the small sample size. Nevertheless, we assume that it might have importance. According to Devine et al., the asymmetry in ALS is important; their results showed that limb dominance plays an important role and may reflect underlying neuronal vulnerabilities that become exposed by the disease. Additionally, they found that the onset of weakness was more likely to occur in the dominant upper limb [40]. In a critical review, a poorer prognosis with lower limb onset was shown, as the authors assumed, due to an increased risk of thromboembolic disease and infections arising from loss of motility [41]. However, in our study, the functional score was lower in the right lower limb group, where the BR answers were also increased significantly compared to the right upper limb. This might mean that the overall severity of the disease was worse, and that not only the secondary causes may be responsible for more severe outcomes. Undoubtedly, disturbances in writing, speech, and swallowing are more alarming symptoms than a unilateral mild paresis in the lower limb, and these patients recognized their symptoms earlier than in the lower-limb-onset groups. At this time, the BR abnormalities and ASLFRS-R score were already higher (Figure 7).

Limitations of this study include the relatively small number of individuals with ALS (*N* = 29), while we enrolled 50 healthy controls and 61 patients with other diseases. However, in BR studies, usually 12–25 patients are enrolled with 20–25 control patients [10,11,25,26]. We also had to deal with the COVID-19 pandemic within the studied time interval, which made both the diagnosis of potential patients and the access and examination of already-diagnosed ALS patients more difficult, while also disrupting our research processes. Nevertheless, these patients were examined within one center and one region, creating a homogeneous study population. Another advantage might be that more patient groups were tested in addition to healthy controls for comparison, providing the opportunity for more precise interpretation. The rarity of the disease and the limited number of cases make it necessary to follow up patients for a longer period of time and sustain the study on a larger sample of patients diagnosed with ALS.

Adequate counselling and better care could be achieved not only with an early diagnosis of ALS, but also with a good monitoring biomarker. The examination and follow-up of patients with suspected ALS can be difficult for several reasons. The absence of well-equipped neurophysiological laboratories and MRI can limit examinations, and not only the diagnosis but also precise follow-up may be delayed, leading to less effective care. In contrast, the BR examination is inexpensive and reliable, and reproducible results can be obtained. Moreover, it is a painless, non-invasive diagnostic method that is easy to use and does not involve long diagnostic procedures. According to our results, BR examination might be a promising tool to monitor the course of the disease and to serve as a biomarker in patients with ALS, due to its correlation with ALSFRS-R, but this should be assessed in further studies. BR might be of prognostic or pathophysiological interest to study in ALS. Therefore, neurophysiological examinations, including BR, may play a more pronounced role in future clinical trials for ALS [38].

## 5. Conclusions

In conclusion, a BR examination is a fast and inexpensive tool to study ALS. It might be a promising tool to monitor the course of the disease or to serve as a prognostic biomarker in patients with ALS, but this should be assessed in further studies. In this way, BR might help with earlier interventions in order to maintain a better functional status in ALS patients.

## Figures and Tables

**Figure 1 brainsci-13-01384-f001:**
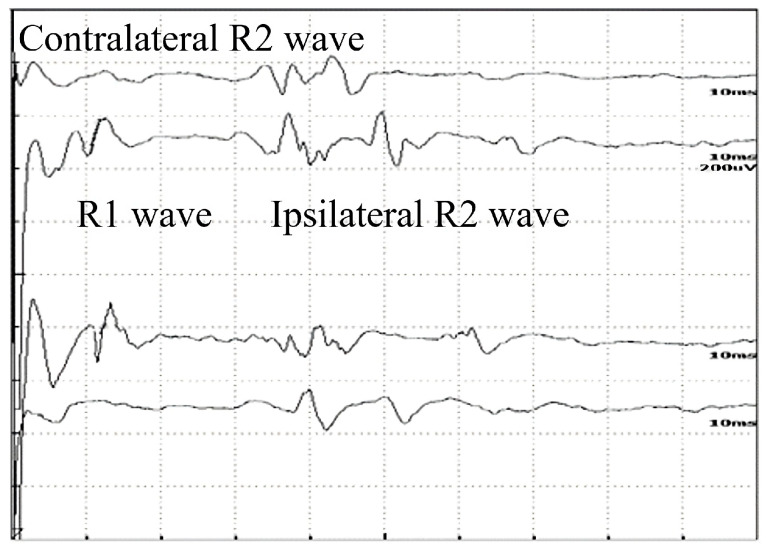
Normal record of the blink reflex waves of a 32-year-old healthy woman.

**Figure 2 brainsci-13-01384-f002:**
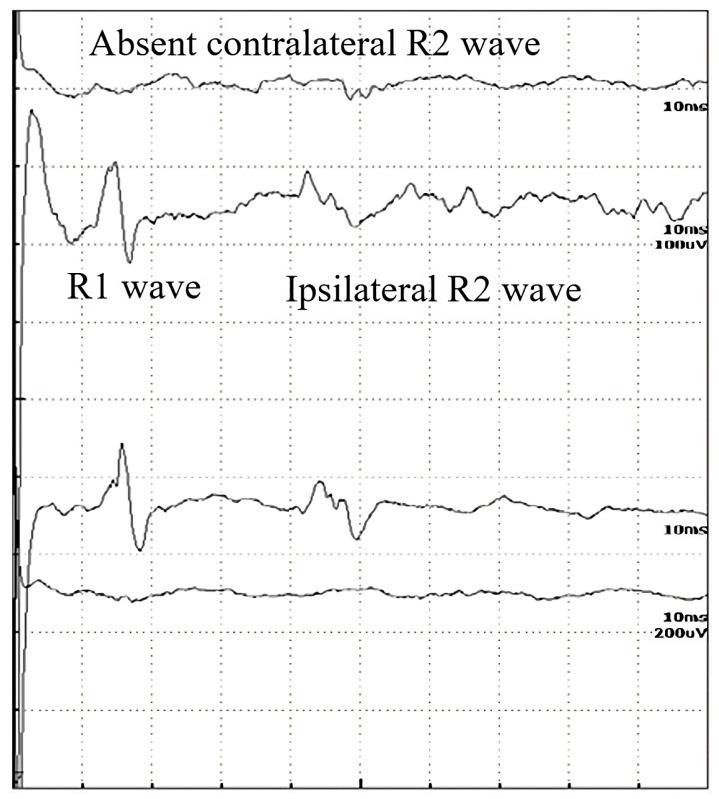
Record of the BR study of a 56-year-old male patient with a disappeared contralateral R2 wave, in the late stages of ALS.

**Figure 3 brainsci-13-01384-f003:**
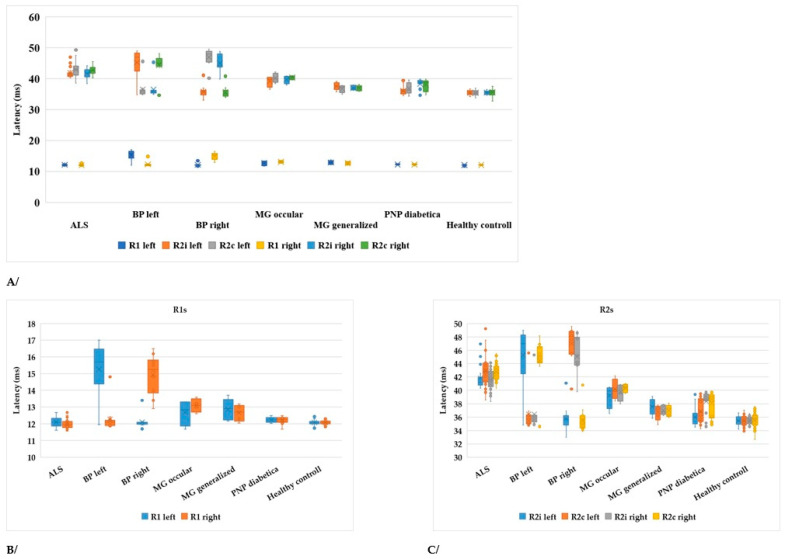
(**A**) Latencies of R1, R2i, and R2c in different diseases of the peripheral nervous system and healthy controls compared to ALS (ALS: amyotrophic lateral sclerosis, BP left: Bell’s paresis left side, BP right: Bell’s paresis right side, MG: myasthenia gravis, PNP: polyneuropathy). (**B**) R1 values in different diseases (axis changed). (**C**) R2 values (axis changed).

**Figure 4 brainsci-13-01384-f004:**
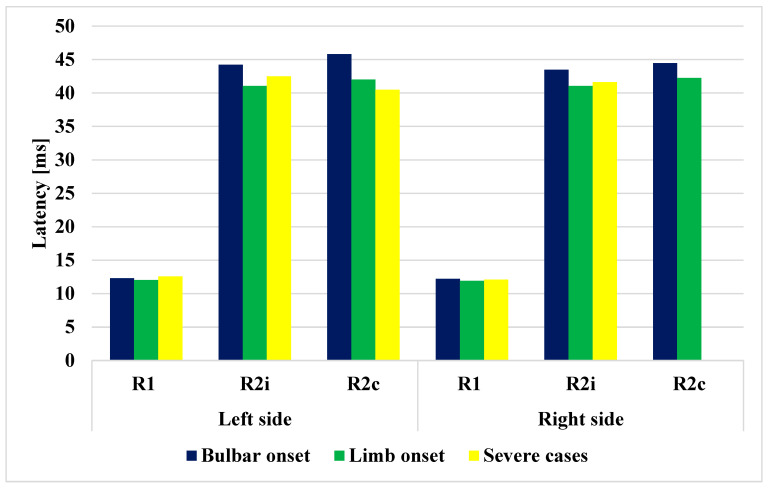
Comparison of latencies of R1, R2i, and R2c in the different ALS subgroups.

**Figure 5 brainsci-13-01384-f005:**
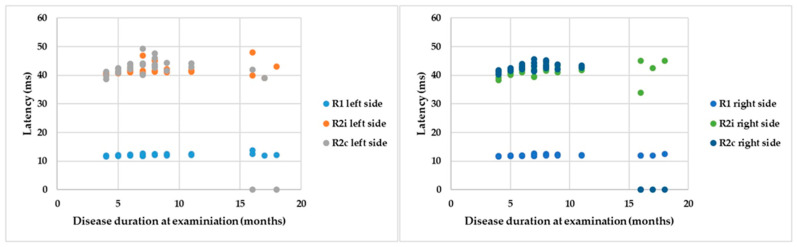
Parameters of BR responses and disease duration.

**Figure 6 brainsci-13-01384-f006:**
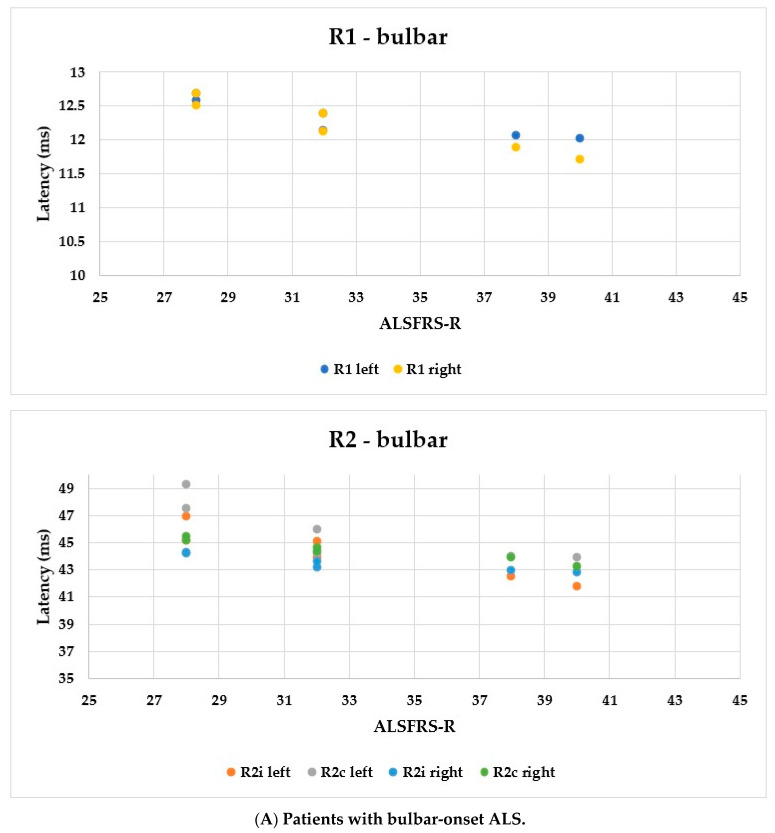
Latencies of R1, R2i, and R2c in patients with ALS and ALSFRS-R.

**Figure 7 brainsci-13-01384-f007:**
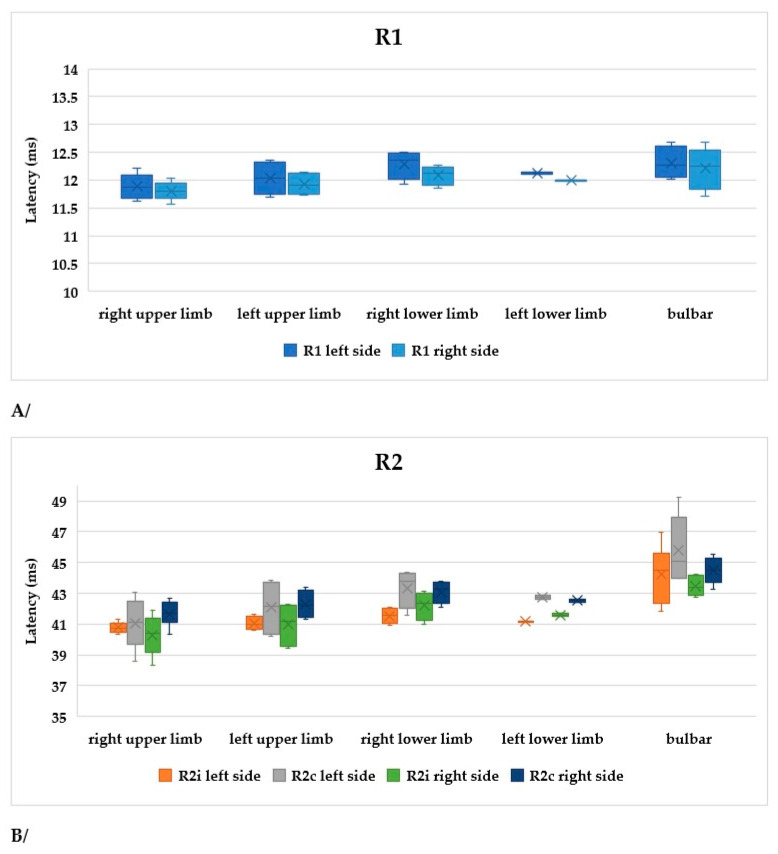
(**A**) Latencies of R1 in patients with ALS. (**B**) Latencies of R2i and R2c in patients with ALS (right and left side, upper and lower onset). (**C**) Latencies depending on ALSFRS-R scores.

**Table 1 brainsci-13-01384-t001:** Patients’ characteristics.

	All Patients
Patient number	29
Female/male (number)	12/17
Age (years) (mean ± SD)	67.86 ± 10.21
Disease duration (months) (mean ± SD)	8.21 ± 3.93
ALSFRS-R (mean ± SD)	33.13 ± 6.72

**Table 2 brainsci-13-01384-t002:** Latencies of R1, R2i, and R2c in the healthy control group and in ALS patients compared to other diseases: (A) Latencies of R1, R2i, and R2c in the healthy control group, in ALS patients, and in different diseases of the peripheral nervous system. (B) The *p*-values of latencies of R1, R2i, and R2c in different diseases of the peripheral nervous system and ALS patients compared to healthy controls. (C) The *p*-values of latencies of R1, R2i, and R2c in different diseases of the peripheral nervous system compared to ALS patients.

(A)																				
				ALS			Bell’s Paresis, Left Side		Bell’s Paresis, Right Side			Myasthenia Gravis, Ocular			Myasthenia Gravis, Generalized			Diabetic PNP		Healthy Controls
Patient number				29			12		15			4			5			25		50
Age (years) (mean ± SD)				67.86 ± 10.21			54 ± 13.65		52.53 ± 15.42			53.5 ± 13.6			63 ± 18.53			68.24 ± 10.64		52.94 ± 17.74
Left side	R1			12.17 ± 0.4			15.63 ± 1.1		12.01 ± 0.1			13.1 ± 0.47			12.76 ± 0.78			12.23 ± 0.15		12.07 ± 0.14
	R2i			41.93 ± 2.05			46.58 ± 2.62 +3 absent		35.37 ± 0.99			40.35 ± 0.6			37.04 ± 0.93			36.08 ± 1.34		35.49 ± 0.66
	R2c			42.76 ± 2.41 +2 absent			35.65 ± 0.68		47.54 ± 1.58 +5 absent			40.63 ± 1.05			37.44 ± 0.81			36.86 ± 1.73		35.41 ± 0.8
Right side	R1			12.0 ± 0.26			12.06 ± 0.17		15.14 ± 0.94 +5 absent			13.18 ± 0.31			12.76 ± 0.35			12.23 ± 0.15		12.06 ± 0.13
	R2i			41.63 ± 2.25			35.69 ± 0.51		45.59 ± 2.03 +5 absent			40.18 ± 0.69			37.4 ± 0.65			38.42 ± 1.6		35.48 ± 0.53
	R2c			42.78 ± 1.3 +4 absent			45.56 ± 1.4 +3 absent		35.14 ± 0.92			40.38 ± 0.68			37.94 ± 1.67			38.1 ± 1.75		35.55 ± 1.03
(**B**)																				
					**ALS**			**Diabetic Polyneuropathy**			**Myasthenia Gravis, Generalized**			**Myasthenia Gravis, Ocular**			**Bell’s Paresis, Right Side**			**Bell’s Paresis, Left Side**
Healthy controls		R1 left			0.12			**<0.0001**			**<0.0001**			**<0.0001**			0.13			**<0.0001**
		R2i left			**<0.0001**			**0.01**			**<0.0001**			**<0.0001**			0.59			**<0.0001**
		R2c left			**<0.0001**			**<0.0001**			**<0.0001**			**<0.0001**			**<0.0001**			0.34
		R1 right			0.18			**<0.0001**			**<0.0001**			**<0.0001**			**<0.0001**			0.95
		R2i right			**<0.0001**			**<0.0001**			**<0.0001**			**<0.0001**			**<0.0001**			0.22
		R2c right			**<0.0001**			**<0.0001**			**<0.0001**			**<0.0001**			0.17			**<0.0001**
(**C**)																				
						**Diabetic Polyneuropathy**				**Myasthenia Gravis, Generalized**			**Myasthenia Gravis, Ocular**			**Bell’s Paresis, Right Side**			**Bell’s Paresis, Left Side**	
ALS			R1 left			0.48 *				**0.01 ***			**0.0001 ***			0.13			**<0.0001 ***	
			R2i left			**<0.0001**				**<0.0001**			0.14			**<0.0001**			**<0.0001 ***	
			R2c left			**<0.0001**				**<0.0001**			0.096			**<0.0001 ***			**<0.0001**	
			R1 right			**0.0003 ***				**<0.0001 ***			**<0.0001 ***			**<0.0001 ***			0.47 *	
			R2i right			**<0.0001**				**0.0002**			0.21			**<0.0001 ***			**<0.0001**	
			R2c right			**<0.0001**				**<0.0001**			**0.0013**			**<0.0001**			**<0.0001 ***	

Bold: statistically significant; * latency is shorter in ALS.

**Table 3 brainsci-13-01384-t003:** Characteristics and findings of the subgroups; *p*-values corresponding to bulbar onset (b), limb onset (l), and severe cases (s).

		Bulbar Onset	*p (b-l)*	*p (b-s)*	Limb Onset	*p (l-s)*	Severe Cases
Patient number		6			19		4
Female/male (number)		4/2			6/13		2/2
Age (years) (mean ± SD)		57.5 ± 13.91	0.004	0.099	70.32 ± 6.98	0.72	71.75 ± 9.25
Disease duration (months) (mean ± SD)		7 ± 0.89	0.82	<0.0001	6.79 ± 2.2	<0.0001	16.75 ± 0.96
Left side	R1	12.31 ± 0.28	0.04	0.45	12.04 ± 0.26	0.02	12.58 ± 0.78
	R2i	44.24 ± 1.88	<0.0001	0.38	41.07 ± 0.45	0.12	42.5 ± 4.04
	R2c	45.82 ± 2.2	0.0001	0.005	42.03 ± 1.6	0.11	40.5 ± 2.12
Right side	R1	12.21 ± 0.38	0.016	0.56	11.92 ± 0.18	0.13	12.08 ± 0.22
	R2i	43.48 ± 0.63	0.0002	0.4	41.05 ± 1.27	0.65	41.63 ± 5.29
	R2c	44.48 ± 0.83	<0.0001	N/A	42.25 ± 0.89	N/A	0
ALSFRS-R (mean ± SD)		33 ± 5.02	0.24	0.0017	35.74 ± 4.83	<0.0001	21.0 ± 0.82

**Table 4 brainsci-13-01384-t004:** Characteristics and findings of ALS patients with limb onset (ru: right upper limb, lu: left upper limb, rl: right lower limb, ll: left lower limb).

		Right Upper Limb	*p* *(ru-lu)*	*p* *(ru-rl)*	*p* *(ru-ll)*	Left Upper Limb	*p* *(lu-rl)*	*p* *(lu-ll)*	Right Lower Limb	*p* *(rl-ll)*	Left Lower Limb
Patient number		7				6			4		2
Age (years) (mean ± SD)		69.14 ± 7.8	0.79	0.98	0.27	70.17 ± 5.46	0.83	0.24	69.25 ± 7.37	0.33	77 ± 9.9
Disease duration until examination (months) (mean ± SD)		5.43 ± 1.72	0.22	0.02	0.44	7 ± 2.61	0.35	0.88	8.5 ± 1.73	0.78	7.5 ± 0.71
Left side	R1	11.89 ± 0.21	0.32	0.02	0.17	12.03 ± 0.27	0.17	0.64	12.29 ± 0.26	0.46	12.13 ± 0.02
	R2i	40.78 ± 0.34	0.18	0.014	0.16	41.07 ± 0.4	0.14	0.72	41.54 ± 0.5	0.39	41.18 ± 0.05
	R2c	41.03 ± 1.53	0.25	0.03	0.18	42.07 ± 1.56	0.21	0.59	43.36 ± 1.27	0.55	42.73 ± 0.16
Right side	R1	11.81 ± 0.16	0.26	0.02	0.17	11.92 ± 0.17	0.16	0.6	12.09 ± 0.17	0.48	11.99 ± 0.01
	R2i	40.29 ± 1.23	0.33	0.024	0.19	40.98 ± 1.22	0.12	0.52	42.22 ± 0.91	0.42	41.6 ± 0.13
	R2c	41.65 ± 0.79	0.18	0.015	0.18	42.29 ± 0.83	0.16	0.72	43.09 ± 0.72	0.35	42.52 ± 0.13
ALSFRS-R (mean ± SD)		38.86 ± 3.8	0.25	0.013	N/C	35.83 ± 5.19	0.19	N/C	31.5 ± 3.79	N/C	33 ± 0

N/C: not computable.

## Data Availability

The data presented in this study are available on request from the corresponding author. The data are not publicly available due to ethical reasons.

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
