# Peer review of "Blink Reflex Examination in Patients with Amyotrophic Lateral Sclerosis Compared to Diseases Affecting the Peripheral Nervous System and Healthy Controls"

_brainsci, 2023, doi:10.3390/brainsci13101384_

Round 1

Reviewer 1 Report

The study investigated Blink Reflex abnormalities in oligo- and poly-synaptic pathways in patients with ALS, in comparison to both healthy controls and a variety of other neurological condition involving the peripheral nervous system. However, these conditions are only occasionally in differential diagnosis with motor neuron disease and thus the utility of this comparison is limited.

ALS patients demonstrated an increased latency in particular in poly-sinaptic responses that hint at a central bulbar involvement also in patients with limb-onset ALS, even though it is not stated whether they had overt bulbar involvement at the time of evaluation. To my better knowledge, the finding is novel and, if confirmed, could be useful as an inexpensive and easily obtainable neurophysiological evidence of bulbar involvement.

I find the overall study interesting, even though there are several issues that need to be addressed:

- Introduction is hypertrophic. Please consider shortening considerably both the part on the ALS and the BR background.

- Methods: the choice to test a single muscle and in particular the extensor digitorum communis to serve as an index of overall motor involvement is peculiar and not supported by other studies (the authors provide as citations [30] a study investigating radial palsy by assessment of three muscle groups, not ALS and [29] a study investigating CIDP patients using the Neurological Disease Score, assessing 26 muscle groups). As ALS often begins with a focal predominance, this measure seems insufficient as a measure even for overall limb motor impairment. I would either provide a MRC sum score or amend the measure completely

- Statistical analysis: comparisons between multiple groups (three groups when assessing ALS, six when healthy and pathologic controls are included) should be assessed first with Kruscal-Wallis or ANOVA, proceeding to Mann-Whitney and t-test only if the former are significant, adopting a p value corrected for the multiple comparisons.

- Results and Discussion: the study is about BR in ALS. I would try to maintain the focus on this and not digress exceedingly with the findings on other conditions.

- Discussion: needs improvement. The main results of their study start at lines 322, after 40 lines and 7 paragraphs of introduction and consideration on the sample, try to rephrase and leave unnecessary information out in order to better highlight these findings.

- Discussion, line 322-323: BR pathological values usually means that they scored outside the 95° percentile or 95% CI for healthy controls, and the authors stated that all their ALS patients had pathological values. Even though the data provided hint at this, the share of patients with pathological values is never formally reported in the results. From a clinical point of view, a difference observable at patient-level would be of greater interest compared to a difference on a group-level only.

- Discussion: the finding on the R2 latency and R2c disappearance in ALS seems to be the strongest point of the study. The authors aptly postulated an involvement of interneurons. Did the authors find any neuropathology study with similar results?

Minor points

- line 62: I would rephrase the period in order to convey both that the neurophysiology is required since the El Escorial criteria AND maintained also in the most recent sets of criteria

- line 103: please change "BR answers" to "BR responses" for consistency with the remaining text

- line 191-193: I would use "show extreme variability" or similar terms rather than "show variability"

- line 197-198: did the patients with absent R2c have a normal R1 on that side? on the same matter, having excluded these patients from the analysis and then stating that absent responses are important seems conflicting

- during the entire Results section, please be consistent when referring to "controls" if they are considered to be Healthy Controls only or All Controls, including those with other diseases as well

- line 213-214: the finding of longer R2 in MG is unexpected and, to my better knowledge, not described before. Do the authors have found other studies and how they interpret this finding?

- Table I: check the cell of absent R2 response in the Bell-paresis left side group

- Figure 3: two separate graphs for R1s and R2s seems a better option, as it is difficult to see the R1 variability in this

- Figures: please use a similar scale for similar studies throughout the test for better comparability (ie Latency from 0 to 60ms)

Needs proofreading.

Author Response

Dear Reviewer,

we are grateful for the valuable comments and remarks we have revised the manuscript according to them.

Comment:

The study investigated Blink Reflex abnormalities in oligo- and poly-synaptic pathways in patients with ALS, in comparison to both healthy controls and a variety of other neurological condition involving the peripheral nervous system. However, these conditions are only occasionally in differential diagnosis with motor neuron disease and thus the utility of this comparison is limited.

ALS patients demonstrated an increased latency in particular in poly-sinaptic responses that hint at a central bulbar involvement also in patients with limb-onset ALS, even though it is not stated whether they had overt bulbar involvement at the time of evaluation. To my better knowledge, the finding is novel and, if confirmed, could be useful as an inexpensive and easily obtainable neurophysiological evidence of bulbar involvement. I find the overall study interesting, even though there are several issues that need to be addressed:

Answer: We are grateful for the valuable comments and remarks we have revised the manuscript according to them. Afterwards an English lecturer supervised the manuscript.

Comment:

- Introduction is hypertrophic. Please consider shortening considerably both the part on the ALS and the BR background.

Answer:

Thank you for the comment, we have shortened it.

Comment:

- Methods: the choice to test a single muscle and in particular the extensor digitorum communis to serve as an index of overall motor involvement is peculiar and not supported by other studies (the authors provide as citations [30] a study investigating radial palsy by assessment of three muscle groups, not ALS and [29] a study investigating CIDP patients using the Neurological Disease Score, assessing 26 muscle groups). As ALS often begins with a focal predominance, this measure seems insufficient as a measure even for overall limb motor impairment. I would either provide a MRC sum score or amend the measure completely

Answer:

We aimed to find a relatively spared muscle, but you are right, an MRC sum score may indeed be better. Nevertheless ALSFRS-R gives a clearer picture of the patient. As advised, we left out the parts concerning MRC in EDC muscle and modified the manuscript as follows. Additionally, the original Figure 5 was deleted and the original Table 2 and 3 modified.

Abstract:

ALS functional rating scale - revised (ALSFRS-R) was used to evaluate functional status.”

Methods

“Amyotrophic Lateral Sclerosis Functional Rating Scale – Revised (ALSFRS-R) score [28, 29] was used to assess functional status.”

Results:

“If muscle strength was examined between the groups, the severe group showed the worst results compared to the limb- and bulbar-onset groups. These data show that at the time of the BR examination - except for the severe group - our patients were in the mild or moderate stage from the point of view of muscle strength. Bulbar- and limb-onset groups did not show significant differences in ALSFRS-R (Table 2).

If the different subgroups of ALS are examined (bulbar, limb) significant differences can be observed in terms of age (p=0.004). The latencies of R1, R2i, R2c are all increased in the bulbar subtype compared to the limb-onset subtype (p<0.05). The four severe cases had worse ALSFRS-R scores and disease duration before examination was also significantly longer. (p<0.0001).”

Comment:

- Statistical analysis: comparisons between multiple groups (three groups when assessing ALS, six when healthy and pathologic controls are included) should be assessed first with Kruscal-Wallis or ANOVA, proceeding to Mann-Whitney and t-test only if the former are significant, adopting a p value corrected for the multiple comparisons.

Answer:

Thank you for the comment. We added the following paragraph to clarify the text:

“Normality analyses were conducted: in case the distribution was normal we carried out ANOVA and if the distribution was not normal Kruskal-Wallis analysis was done by multiple comparisons.  When comparing two groups, the Mann-Whitney test for non-parametric and the t-test as a parametric test were conducted. For age comparison either the t-test or ANOVA was used and the other parameters were tested with Kruskal-Wallis or Mann-Whitney analysis.”

Comment:

- Results and Discussion: the study is about BR in ALS. I would try to maintain the focus on this and not digress exceedingly with the findings on other conditions.

Answer:

Thank you for the comment. We hope that the main goal of our work is emphasized better, and could show that we used these diseases’ BR patterns only to understand ALS pathomechanism better and tried to shorten and rephrase parts.

“Trying to understand the pathomechanism of BR in ALS patients we found an interesting aspect of BR parameters in a peripheral nerve in diabetic polyneuropathy without any sign of cranial nerve dysfunction.”

The reference was enriched with the following publications:

ŠtÄ›tkáÅ™ová I, Ehler E. Diagnostics of Amyotrophic Lateral Sclerosis: Up to Date. Diagnostics (Basel). 2021 Feb 3;11(2):231. doi: 10.3390/diagnostics11020231

Aramideh M, Ongerboer de Visser BW, Koelman JH, Majoie CB, Holstege G. The late blink reflex response abnormality due to lesion of the lateral tegmental field. Brain (1997); 120: 1685–92.

Ongerboer de Visser BW, Kuypers HG. Late blink reflex changes in lateral medullary lesions. An electrophysiological and neuro-anatomical study of Wallenberg's syndrome. Brain1978; 101: 285–94.

Tamai Y, Iwamoto M, Tsujimoto T. Pathway of the blink reflex in the brainstem of the cat: interneurons between the trigeminal nuclei and the facial nucleus. Brain Research. (1986) 13;380(1):19-25. doi: 10.1016/0006-8993(86)91424-1

Pars were rephrased in the Discussion as well, please see the answers in the next comment’s answer.

Comment:

- Discussion: needs improvement. The main results of their study start at lines 322, after 40 lines and 7 paragraphs of introduction and consideration on the sample, try to rephrase and leave unnecessary information out in order to better highlight these findings.

Answer:

Thank you for the comment, we have shortened the introduction. We did our best to highlight the necessary information and rephrased sentences in the Discussion.

“The diagnosis of ALS is still challenging, especially in the earlier stages of the dis-ease. EMG and other neurophysiological studies are important diagnostic methods for all patients who present with clinical signs of ALS [35, 37, and 38]. Usually, blinking and other upper facial movements can be retained until the late stage of ALS, making blink reflex a potential supportive tool. Besides this not only the conduction of the re-flex arch, but also the structures of the affected brainstem can be examined using BR studies. However, there are some studies [10, 11, 25, 26, and 27] that highlight the im-portance of BR in the assessment of the pathophysiology of ALS.”

“An early diagnosis of ALS would allow more adequate counselling, better care. Due to several conditions, the examination of patients with suspected ALS could be difficult. The absence of well-equipped neurophysiological laboratories and MRI can limit examinations and diagnosis may also be delayed. In contrast, the BR examination is inexpensive and reliable, and reproducible results can be obtained. Moreover, it is a painless, non-invasive diagnostic method, which is easy to use and does not involve long diagnostic procedures. Therefore, BR examination may help to support the suspicion of the diagnosis of ALS and to distinguish from other disease and give earlier in-sight in the disease status then the clinical examination alone according to our findings. Neurophysiological examinations may play a role in the future in clinical trials of ALS more pronounced [42], among them BR might be also an important option in our opinion.”

“Of course, some limitations occur by these comparisons. The character of the diseases sometimes already determines the age as well, so e.g. in myasthenia gravis, patients were younger. Nevertheless, the pathological latencies were increased and showed a pattern that could be well distinguished from the ‘ALS pattern’.  “

“By examining BR responses in the light of disease duration the correlation has to be in-terpreted very cautiously. Although p values might be significant, but accompanied with a less powerful ρ value. This might be either due to small sample size, and/ or the characteristic of the course of the disease, which might have great interindividual differences. Trying to understand the pathomechanism of BR in ALS patients we found an interesting aspect of BR parameters in a peripheral nerve in diabetic polyneuropathy without any sign of cranial nerve dysfunction.”

Comment:

- Discussion, line 322-323: BR pathological values usually means that they scored outside the 95° percentile or 95% CI for healthy controls, and the authors stated that all their ALS patients had pathological values. Even though the data provided hint at this, the share of patients with pathological values is never formally reported in the results. From a clinical point of view, a difference observable at patient-level would be of greater interest compared to a difference on a group-level only.

Answer:

We presented all data in Table 1 by comparing healthy controls and diseased controls as well. We added to the Methods:

“BR parameters were considered pathological if the value was outside the 95° percentile for healthy controls. The values are the following: R1 left 12.3ms, R2i 36.44ms, R2c 36.65ms, R1 right 12.28ms, R2c 36.28ms, R2c 37.13ms.”

Comment:

- Discussion: the finding on the R2 latency and R2c disappearance in ALS seems to be the strongest point of the study. The authors aptly postulated an involvement of interneurons. Did the authors find any neuropathology study with similar results?

Answer:

Thank you, we have added the following sentence:

“Interneurons play a crucial role in mediating the R2 wave of the BR, as support by several studies, including neuropathological research, that corroborated similar findings. [30,31,32]”

Comment:

Minor points

- line 62: I would rephrase the period in order to convey both that the neurophysiology is required since the El Escorial criteria AND maintained also in the most recent sets of criteria

Answer:

It has been rephrased.

“Neurophysiology has been incorporated into the El Escorial criteria and is essential in the diagnosis of ALS.”

Comment:

- line 103: please change "BR answers" to "BR responses" for consistency with the remaining text

Answer:

It has been changed.

Comment:

- line 191-193: I would use "show extreme variability" or similar terms rather than "show variability"

Answer:

It has been changed accordingly.

“Figure 1 demonstrates a normal record. R1 and R2 responses (both ipsi-and contralateral) also show extreme variability in amplitude in the normal populations.”

Comment:

- line 197-198: did the patients with absent R2c have a normal R1 on that side? on the same matter, having excluded these patients from the analysis and then stating that absent responses are important seems conflicting

Answer:

Thank you for the comment. R1 could be elicited, but R2c not. Since we report on these findings in the tables and results, we corrected the part as follows:

“Four patients with ALS were hospitalised at a later stage of the disease. Their contralateral R2 waves had completely disappeared and could not therefore be quantified (Figure 2). The absent responses are also important to study. In ALS patients, the absent R2c responses may show the importance of interneurons.”

Comment:

- during the entire Results section, please be consistent when referring to "controls" if they are considered to be Healthy Controls only or All Controls, including those with other diseases as well

Answer:

Thank you. Wherever it was missing, we added the term.

Comment:

- line 213-214: the finding of longer R2 in MG is unexpected and, to my better knowledge, not described before. Do the authors have found other studies and how they interpret this finding?

Answer:

It might be due to our patient selection. Our subjects were patients with newly discovered myasthenia gravis. This might be important because we think that habituation might be responsible for the finding as in a publication of Shahani:

“It was noted that low frequency stimulation resulted in increased latency and diminution in the amplitude of the second component (habituation), the delayed component on the contralateral side being most susceptible.”

Shahani B. The human blink reflex. J Neurol Neurosurg Psychiatry. 1970 Dec;33(6):792-800. doi: 10.1136/jnnp.33.6.792.

Comment:

- Table I: check the cell of absent R2 response in the Bell-paresis left side group

Answer:

Thank you. We have corrected it, it was misplaced.

Comment:

- Figure 3: two separate graphs for R1s and R2s seems a better option, as it is difficult to see the R1 variability in this

Answer:

The graphs in Fig. 3 have been separated.

Figure 3 A/Latencies of R1, R2i, R2c in different disease of the peripheral nervous system and healthy controls compared to ALS. (ALS: amyotrophic lateral sclerosis, BP left: Bell paresis left side, BP right: Bell paresis right side, MG: myasthenia gravis, PNP: polyneuropathy) B/ R1 values in different diseases (axis changed) C/ R2 values (axis changed).

Comment:

- Figures: please use a similar scale for similar studies throughout the test for better comparability (ie Latency from 0 to 60ms)

Answer:

We changed the figures as recommended. Our aim was to enhance data visibly. 

“Figure 3 Latencies of R1, R2i, R2c in different diseases of the peripheral nervous system and healthy controls compared to ALS. (ALS: amyotrophic lateral sclerosis, BP left: Bell paresis left side, BP right: Bell paresis right side, MG: myasthenia gravis, PNP: polyneuropathy)”

Reviewer 2 Report

Rostàs et al. conducted a study on the diagnostic and prognostic value of the blink reflex in ALS patients, comparing R1 and R2 latencies of ALS patients with healthy controls’ and other diseases' groups which can show blink reflex abnormalities.

Although the study is interesting, I have some concerns and suggestions:

- In the introduction the authors stated that patients with a definite diagnosis of ALS were enrolled. However, in the method section, the authors stated that patients with UMN and LMN dysfunctions in at least one body region were recruited (reflecting a diagnosis of at least possible ALS and not definite ALS). Please specify in the manuscript the inclusion criteria for participating in the study and cite the criteria used for ALS diagnosis.

- I do not understand why the authors chose the EDC muscle to examine the MRC. I would have preferred a muscle innervated by the facial nerve (mainly the orbicularis oculi) in order to allow some neurophysiological-clinical correlations. Furthermore, being ALS a focal disease in the early stages, ECD is not representative of the disease severity.

- Although the disease groups chosen as “disease controls” are represented by diseases showing blink reflex abnormalities, some of them are not “ALS mimics”, such as Bell paresis or diabetic polyneuropathy. I would have preferred some diseases which actually can challenge the diagnosis, such as MMN, HSP, hirayama, etc..

- Beside the heathy controls group, also the MG and the Bell paresis groups are younger than ALS patients, and this might influence the results of the study.

- In the statistical analysis, both non parametric (Mann-Whitney test) and parametric studies (T-test) are mentioned. Could the authors specify when did they use one or the other test and why?

- In the first part of results, please specify: how many patients with bulbar onset, how many patients with limb onset, the disease duration (months from the disease onset to blink reflex examination)

- Although the authors presented the correlation between ALSFRS and latencies with a qualitative method (for example figure 6), it would be more appropriate to use a statistical method (for example, the Spearman’s correlation). It would be also interesting to explore the correlation between the R latencies and the disease duration, since the most severe patients did not have R2 responses (suggesting a correlation with the advanced stage of the disease).

- The “prognostic value” of the blink reflex is poorly explored (only the qualitative association with ALSFRS). I would not emphasize the “prognostic value”, but the study of “neurophysiological-clinical correlations”.  

- In the figures please do not use suspension dots, but abbreviations (as in the tables).

- In the abstract, line 26, correct “disease” with “diseases”

Author Response

Dear Reviewer,

we are grateful for the valuable comments and remarks we have revised the manuscript according to them.

Comment:

Rostàs et al. conducted a study on the diagnostic and prognostic value of the blink reflex in ALS patients, comparing R1 and R2 latencies of ALS patients with healthy controls’ and other diseases' groups which can show blink reflex abnormalities.

Although the study is interesting, I have some concerns and suggestions:

Answer:

We are grateful for the valuable comments and remarks we have revised the manuscript according to them. Afterwards an English lecturer supervised the manuscript.

Comment:

- In the introduction the authors stated that patients with a definite diagnosis of ALS were enrolled. However, in the method section, the authors stated that patients with UMN and LMN dysfunctions in at least one body region were recruited (reflecting a diagnosis of at least possible ALS and not definite ALS). Please specify in the manuscript the inclusion criteria for participating in the study and cite the criteria used for ALS diagnosis.

Answer:

The patients had definitive ALS according to the El Escorial (3 limbs), so we have modified the text accordingly.

“All of the patients had had clinical signs of UMN and LMN dysfunction in three regions revealed by repeated clinical assessments [8]. The definite diagnosis of ALS and willing to take part in the examinations served as inclusion criteria.”

Comment:

- I do not understand why the authors chose the EDC muscle to examine the MRC. I would have preferred a muscle innervated by the facial nerve (mainly the orbicularis oculi) in order to allow some neurophysiological-clinical correlations. Furthermore, being ALS a focal disease in the early stages, ECD is not representative of the disease severity.

Answer:

Both the orbicularis oculi and frontalis muscles were examined. By the examined patients, both were without clinical findings. The facial muscles were intact in these patients. We aimed to find a relatively spared muscle. The EDC muscle was chosen because all patients could innervate it, and further studies could be carried out. ALSFRS-R enabled us to refine the overall picture, so we left out the parts concerning MRC in EDC muscle and modified the manuscript as follows. Additionally, the original Figure 5 was deleted and the original Table 2 and 3 modified.

We added:

“By the examined patients, orbicularis oculi and frontalis muscle were without clinical findings.”

We modified:

Abstract:

ALS functional rating scale - revised (ALSFRS-R) was used to evaluate functional status.”

Methods

“Amyotrophic Lateral Sclerosis Functional Rating Scale – Revised (ALSFRS-R) score [28, 29] was used to assess functional status.”

Results:

“If muscle strength was examined between the groups, the severe group showed the worst results compared to the limb- and bulbar-onset groups. These data show that at the time of the BR examination - except for the severe group - our patients were in the mild or moderate stage from the point of view of muscle strength. Bulbar- and limb-onset groups did not show significant differences in ALSFRS-R (Table 2).

If the different subgroups of ALS are examined (bulbar, limb) significant differences can be observed in terms of age (p=0.004). The latencies of R1, R2i, R2c are all increased in the bulbar subtype compared to the limb-onset subtype (p<0.05). The four severe cases had worse ALSFRS-R scores and disease duration before examination was also significantly longer. (p<0.0001).”

Comment:

- Although the disease groups chosen as “disease controls” are represented by diseases showing blink reflex abnormalities, some of them are not “ALS mimics”, such as Bell paresis or diabetic polyneuropathy. I would have preferred some diseases which actually can challenge the diagnosis, such as MMN, HSP, hirayama, etc..

Answer:

Thank you for the comment. We tried to detect these patients, but during the examined period we only had 1 patient with Hirayama and 1 patient with MMN, as these diseases are rare and the size of the population in our catchment area is approximately 600000. Nevertheless, we wanted have disease controls. In the literature, we could not find studies having sufficient number of such patients, which shows the difficulty of recruiting patients with the diseases you have mentioned in the comment.  (Similar studies did not have disease controls either.)

So, we aimed to have controls, which could help to understand at least one side of the pathway to that pathomechanism.

Comment:

- Beside the heathy controls group, also the MG and the Bell paresis groups are younger than ALS patients, and this might influence the results of the study.

Answer:

Sometimes diseases are characterised by an early onset so patients with MG and Bell paresis were younger. Therefore, we focused on the pattern: the latency of R1 was longer in these patients.

We added:

“Of course, these comparisons have some limitations. Sometimes diseases are characterised by an early onset so e.g. in myasthenia gravis, patients were younger. Nevertheless, the pathological latencies were increased and showed a pattern that could be well distinguished from the ‘ALS pattern’.”

Comment:

- In the statistical analysis, both non-parametric (Mann-Whitney test) and parametric studies (T-test) are mentioned. Could the authors specify when did they use one or the other test and why?

Answer:

Thank you for your comment. We added the following paragraph to clarify the text:

“Normality analyses were conducted: in case the distribution was normal we carried out ANOVA and if the distribution was not normal Kruskal-Wallis analysis was done by multiple comparisons.  When comparing two groups, the Mann-Whitney test for non-parametric and the t-test as a parametric test were conducted. For age comparison, either the t-test or ANOVA was used and the other parameters were tested with Kruskal-Wallis or Mann-Whitney analysis.”

Comment:

- In the first part of results, please specify: how many patients with bulbar onset, how many patients with limb onset, the disease duration (months from the disease onset to blink reflex examination)

Answer:

The data were presented in Table 2, but we enriched the text as follows:

“Of the patients 6 had bulbar-onset and 19 limb-onset, the period between the onset of the disease to the time of examination being 7 and 6.79 months, respectively (Table 2).”

Comment:

 Although the authors presented the correlation between ALSFRS and latencies with a qualitative method (for example figure 6), it would be more appropriate to use a statistical method (for example, the Spearman’s correlation). It would be also interesting to explore the correlation between the R latencies and the disease duration, since the most severe patients did not have R2 responses (suggesting a correlation with the advanced stage of the disease).

Answer:

Thank you for the comment. We have added the following sentences:

1.

‘Spearman’s rank correlation was computed to assess the relationship between ALSFRS-R and BR latencies, there was a negative correlation between the variables. In all patients with ALS ρ=-0.95, p<0.0001; ρ =-0.61, p= 0.001; ρ =-0.82 p<0.0001; ρ =-0.90, p<0.0001; ρ =-0.83, p<0.0001; ρ =-0.80, p<0.0001 by the latencies of R1 left, R2i left, R2c left, R1 right, R2i right, R2c right, respectively. Among patients with bulbar-onset ρ =-0.90, p=0.005; ρ =-0.93, p= 0.002; ρ =-0.84 p=0.017; ρ =-0.96, p=0.0008; ρ =-0.94, p=0.002; ρ =-0.97, p=0.0004 by the latencies of R1 left, R2i left, R2c left, R1 right, R2i right, R2c right, respectively. In patients with limb onset ρ =-0.99, p<0.0001; ρ =-0.96, p<0.0001; ρ =-0.98 p<0.0001; ρ =-0.98, p<0.0001; ρ =-0.97, p<0.0001; ρ =-0.97, p<0.0001 by the latencies of R1 left, R2i left, R2c left, R1 right, R2i right, R2c right, respectively.’

  1. “If the R latencies of BR are examined in the light of disease duration from onset until examination, a positive correlation can be detected with Spearman’s rank test: ρ=0.5, p=0.007,ρ=0.18, p= 0.25; ρ=0.02 p=0.39; ρ=0.35, p=0.07; ρ=-0.18, p=0.25; ρ=-0.47, p=0.01 by the latencies of R1 left, R2i left, R2 c left, R1 right, R2i right, R2c right, respectively Fig.5). Despite significant correlations, the r values are low, so cautious interpretation is required.”

Figure 5. has been added

Figure 5 Parameters of BR responses and disease duration

Discussion
“On examining BR responses in the light of disease duration, the correlation has to be in-terpreted very cautiously. Although the p-values might be significant, they are accompanied with a less powerful ρ-value. This might either be due to the small sample size, and/ or the characteristics of the course of the disease, which might show great interindividual differences.”

Comment:

- The “prognostic value” of the blink reflex is poorly explored (only the qualitative association with ALSFRS). I would not emphasize the “prognostic value”, but the study of “neurophysiological-clinical correlations”. 

 Answer:

Thank you, we have refined the part as follows:

“An early diagnosis of ALS would allow more adequate counselling and better care. The examination of patients with suspected ALS can be difficult due to several reasons. The absence of well-equipped neurophysiological laboratories and MRI can limit examinations and diagnosis may also be delayed. In contrast, the BR examination is inexpensive and reliable, and reproducible results can be obtained. Moreover, it is a painless, non-invasive diagnostic method, which is easy to use and does not involve long diagnostic procedures. Therefore, a BR examination may help to confirm the initial diagnosis of ALS. According to our findings, this examination can also help to distinguish ALS from other diseases and give an early insight into the stage of the disease than the clinical examination alone can do. In our opinion, neurophysiological examinations may play a more pronounced role in future clinical trials of ALS [42], in which BR might also be an important option.

“5. Conclusions

In conclusion, a BR examination is a fast and inexpensive tool to support the suspected diagnosis of ALS and, also, follow-up the course of the disease. The abnormalities detected with BR might help an earlier intervention in order to maintain a better functional status in ALS patients.”

Comment:

- In the figures please do not use suspension dots, but abbreviations (as in the tables).

Answer:

Thank you, we corrected them.

Comment:

- In the abstract, line 26, correct “disease” with “diseases”

Thank you, we corrected it.

Round 2

Reviewer 2 Report

The authors addressed all my suggestions and comments. 

I thank them and I believe that the manuscript is now suitable for publication. 

Author Response

Dear Reviewer,

We are grateful for the Reviewer for the suggestions and the evaluation.

Comment:

The authors addressed all my suggestions and comments. 

I thank them and I believe that the manuscript is now suitable for publication. 

Answer:

We are grateful for the Reviewer for the evaluation.